# Shock-Driven Decomposition of Polymers and Polymeric Foams

**DOI:** 10.3390/polym11030493

**Published:** 2019-03-13

**Authors:** Dana M. Dattelbaum, Joshua D. Coe

**Affiliations:** 1Explosives Science and Shock Physics Division, Los Alamos National Laboratory, Los Alamos, NM 87545, USA; 2Theoretical Division, Los Alamos National Laboratory, Los Alamos, NM 87545, USA

**Keywords:** polymers, foams, shock compression, equation of state

## Abstract

Polymers and foams are pervasive in everyday life, as well as in specialized contexts such as space exploration, industry, and defense. They are frequently subject to shock loading in the latter cases, and will chemically decompose to small molecule gases and carbon (soot) under loads of sufficient strength. We review a body of work—most of it performed at Los Alamos National Laboratory—on polymers and foams under extreme conditions. To provide some context, we begin with a brief review of basic concepts in shockwave physics, including features particular to transitions (chemical reaction or phase transition) entailing an abrupt reduction in volume. We then discuss chemical formulations and synthesis, as well as experimental platforms used to interrogate polymers under shock loading. A high-level summary of equations of state for polymers and their decomposition products is provided, and their application illustrated. We then present results including temperatures and product compositions, thresholds for reaction, wave profiles, and some peculiarities of traditional modeling approaches. We close with some thoughts regarding future work.

## 1. Introduction

Polymers, polymeric composites, and polymeric foams are used extensively as cushioning, insulation, structural support, and for shock mitigation. As such, they are frequently exposed to impact or dynamic (high strain rate) loading conditions, their response to which is complex and distinct from that of other material classes. At relatively low stresses (σ≲1 GPa) and up to high strain rates ($\dot{\epsilon }≲10^{7}$/s${}^{-1}$), the volumetric and deviatoric components of their response often are not in equilibrium, and viscoelasticity plays an important role [1,2]. For example, their elastic moduli and strength can be strongly strain rate- and temperature-dependent [2]. Most also undergo order-disorder phase transitions, such as the glass transition separating “glassy” from “rubbery” regimes [3].

Even when complications such as viscoelasticity or phase transitions are neglected, the complexity of polymer response places high demands on equation of state (EOS) models [4,5,6]. The temperature-dependence of their specific heats are seldom reproduced by a single characteristic temperature [7], and porosity introduces the need for compaction or multiphase flow models [8,9,10,11,12,13,14,15,16,17]. Working solutions to each of these problems are discussed below, but our emphasis here will be on an additional complication that has received somewhat less attention: chemical reaction under sufficient load, and shock loading in particular. Polymers decompose chemically above some threshold stress on their principal shock locus (defined in the following subsection) [18,19], and this threshold drops dramatically with increasing initial porosity [6]. We believe and will present evidence that the chemistry involved shares many similarities with that of high explosive decomposition, the obvious differences involving reaction thermicity and (typically) the oxygen balance of the starting material.

The remainder of this Introduction outlines some basic concepts of shock physics, as well as some considerations unique to foams. Section 2 briefly describes the chemical formulation of some representative polymers, then presents the experimental and theoretical methods we and others have used to interrogate these systems. Section 3 illustrates some basic phenomenology of polymers and foams under shock loading, and provides some interpretation. Section 4 concludes and suggests promising avenues for future investigation.

### 1.1. Shock Physics Background

#### 1.1.1. Basic Concepts

Shock waves produce discontinuous changes in material properties while remaining subject to conservation of mass, momentum, and energy across the discontinuity [20,21,22,23,24,25,26,27,28,29]. In addition to standard thermodynamic concepts such as pressure (*P*), volume (*V*), density (ρ=1/V), and energy (*E*), the shock (*U*) and material (commonly referred to as “particle”) velocities (*u*) play an important role and often are the quantities actually measured in an experiment. (Throughout this treatment, all shock velocities are those relative to some inertial “Laboratory” frame: i.e., they are Eulerian.) Properties of unshocked material will be designated by a subscripted ‘0’, those of shocked material by a subscripted ‘H’. For material initially at rest (u=0), the conservation laws applied to singly shocked material are:(1)$$\frac{V_{0}}{V_{H}}=\frac{U}{U-u},$$
(2)$$P_{H}=P_{0}+\rho_{0}Uu,$$
and
(3)$$P_{H}u=\rho_{0}U\left(E_{H}+\frac{1}{2}u^{2}-E_{0}\right).$$
These are also known as the Rankine-Hugoniot relations. If the initial state is known, then (Equation 1)–(Equation 3) constitutes a system of three equations in five unknowns. Substitution and elimination permits direct relation of any three variables, one of the most common (and useful) being elimination of the velocities from (Equation 3):(4)$$E_{H}=E_{0}+\frac{1}{2}P_{H}\left(V_{H}-V_{0}\right).$$
Perhaps more importantly, measurement of any two of the five quantities fully determines the state of the system. The locus of states accessible upon single-shock loading from a fixed origin is known as a *Hugoniot*; if that origin happens to be the ambient state, it is known as the *principal* Hugoniot. It is important to note that Hugoniots are not thermodynamic paths: one does not smoothly evolve a system along a Hugoniot as one would along an isotherm in a diamond anvil cell experiment [30], for instance. Each point of a shock locus represents a discrete transition from the origin. The path actually followed in a given experiment is the chord (known as the Rayleigh line) linking the origin with the shocked state in the *P*-*V* plane. Its slope is found by eliminating *u* from (Equation 1) and (Equation 2),
(5)$$\frac{P_{H}-P_{0}}{V_{0}-V_{H}}={\left(\frac{U}{V_{0}}\right)}^{2}.$$
The condition for shock stability [24,26,27] is that this slope be greater than that of the isentrope passing through the shocked state,
(6)$$-{\left(\frac{\partial P}{\partial V}\right)}_{S,P=P_{H},V=V_{H}}>\frac{P_{H}-P_{0}}{V_{0}-V_{H}}.$$
Pressure and particle velocity must be conserved across a shock interface, giving rise to the vitally important notion of *impedance matching* described more fully in Section 2.2.

In the absence of phase transitions or chemical decomposition, the Hugoniot of many materials is linear in the U−u plane [31]:(7)$$U=c_{0}+su,$$
where the *y*-intercept coincides with the bulk sound velocity, $c_{0}$. The latter quantity is related to the isentropic bulk modulus
(8)$$K_{S}=\rho c_{0}^{2},$$
whose pressure derivative can be derived from the slope *s* through
(9)$$s=(K^{\prime }+1)/4,$$
where
(10)$$K^{\prime }={\left(dK/dP\right)}_{\rho =\rho_{0}}.$$
While (Equation 9) follows from thermodynamic identities and is therefore true of all materials, (Equation 7) is purely phenomenological and typically does not hold for polymers [31]. Their principal Hugoniots are better fit by forms such as
(11)$$U=c_{0}+s_{1}u+s_{2}u^{2},$$
with $s_{2}<0$. The same is true of liquids (the Universal Liquid Hugoniot [32] having been proposed specifically to capture concavity in the shock locus at low particle velocity) and high explosive crystals [33]. What these materials have in common is inhomogeneous electron density at atomistic length scales, in contrast to the simple metals for which (Equation 7) was first proposed. As porosity and disorder are reduced by compression, the material stiffens until its compressibility varies with pressure in a manner consistent with (Equation 7).

#### 1.1.2. Wave Splitting

Shock-driven transitions often are accompanied by abrupt changes in volume. Such changes can be negative (i.e., the material densifies), as in most first-order phase transitions in metals and chemical decomposition of full-density polymers [31], or positive, as in detonating high explosives and chemical decomposition of sufficiently porous foams [6,8]. A few transformations are volume-neutral, such as the chemical dissociation of CCl${}_{4}$ [34,35]. Most solid polymers undergo a densifying transition at P≈20–30 GPa, or u≈2.5–3.5 km/s, on their principal Hugoniot [36]. Its extent can be correlated at least qualitatively with packing efficiency in the original material, which in turn is related to the size and shape of side chain moieties along the polymer backbone. Decomposition of “bulky” chains with strong interchain repulsion frees up much more volume than that of “clean” backbones, resulting in larger reduction upon destruction of long-range order. Shock-driven decomposition of polyethylene (perhaps the cleanest polymer backbone) thus results in volume reduction of 0.5%, less than that of polysulfone or polystyrene by over an order of magnitude (≈15%).

If the volume change is negative, then the shock Rayleigh line (Equation (Equation 5)) crosses the principal Hugoniot at more than one *V*, in violation of the shock stability criterion, Equation (Equation 6). The resultant instability splits the initial wave into two, producing multiwave structures qualitatively such as those illustrated in Figure 1 (after Dremin for KCl [37] and Dattelbaum [35] for benzene). This produces an apparent cusp in the shock locus at A, whose origin we discuss further below. Wave splitting for a densification reaction is characterized by $U_{2}\leq U_{1}$ (the slope of the 2nd wave Rayleigh line is less than or equal to that of the 1st wave), meaning the waves may visibly separate in time if the difference is large enough. This separation produces a delay (the plateau between the two rises) that shortens with increasing strength of the initial shock. For sufficiently strong initial shocks, the transition will be overdriven and appear only as rounding in the front. The risetime of waves following the first reflects the rate of underlying chemical or physical transformations, and their evolution with initial stress is reflective merely of transformation rates that increase with temperature. When rates are high enough for transformation to complete within the risetime of the initial wave (typically ∼1 ns), a single, sharp shock will reappear.

Explosively driven flyer plate experiments underlying the data reported in standard compendia [31,38] measured the arrival at an interface of the first waves *only* shown in Figure 1. This had the effect of placing data points in the “mixed phase” region between loci 1 and 2 of the figure, giving the appearance of a pair of derivative discontinuities in the Hugoniot [21,24,25,26,27,28,34,35,37,39,40,41,42,43,44,45,46]. It is important to note that points in this region do not represent equilibrium thermodynamic data, but rather are artifacts of transformation kinetics and the experimental design.

#### 1.1.3. Shockwave Compression of Porous Foams

At the same mass velocity, porous materials typically have lower wave velocities relative to their fully dense counterparts. Their shock loci often exhibit exaggerated curvature in the U−u plane due to compaction of pores at low particle velocity [6,9,13,31,47,48,49,50]. The combination of lower initial density and wave speed implies lower shock impedance ($\rho_{0}U$), which represents the slope of the Rayleigh line in the P−u plane. All these features are illustrated schematically in Figure 2.

Shock compression of foams readily produces temperatures dramatically exceeding those of solid material shocked to the same final pressure; this is due to the large quantity of energy transferred to the foam in the form of P−V work [6,13,14,16,17,51,52]. Figure 3A,B illustrates Hugoniots for solid and porous material when the initial porosity is not too high. The foam is compressed from an initial specific volume, $V_{00}$, to the solid volume $V_{1}$. Full-density material at volume $V_{0}$ undergoes far less compression to reach the same final pressure at roughly the same final volume. Because internal energy (*E*) is obtained from the integral under the Rayleigh line (the solid gray and dotted areas in Figure 3B), the internal energy and temperature rise are much greater for the foam than for the solid [6,13,14,16,17,51,52].

Variation of initial porosity produces the family of Hugoniot curves depicted in Figure 3C [53]. As porosity increases, the internal energy rise becomes increasingly thermal in nature, and at some point shock heating begins to dominate the compressive response such that it becomes ‘anomalous’. Anomalous Hugoniots turn back on themselves, and P(V) is no longer single-valued. Rising shock pressures yield greater shock volumes, as shown in Figure 3C. This can occur even at modest porosities of 25–50% [53,54]. Anomalous compression has been well-documented in porous tungsten [53], copper [15,16], aluminum, and glass microballoons [55,56]. More recently, we showed that this effect can be convoluted with chemical decomposition in shocked polymer foams such as polyurethane [6].

Anomalous compression has several consequences. The porous Hugoniot does not reach that of the solid (as it does in Figure 3A), as assumed in models such as Snowplow and P-α [17]. This typically results in lower shock velocities than such models would predict. The large internal energy rise can dramatically reduce melt stress on the principal Hugoniot, as well as lower the shock condition for thermally driven processes such as phase transformations and chemical dissociation. In foamed aluminum, the threshold stress for melting can be reduced by 50% at porosities of ∼25–30% [51,53].

The impedance of porous materials increases to a far greater extent upon shock loading than in first shock compression of solids. Increase in the sound velocity at pressure,
(12)$$c_{0}^{2}=-V^{2}{\left(\frac{\partial P}{\partial V}\right)}_{S},$$
results in high rarefaction/release wave velocities in the pre-shocked material, as well as much greater second shock velocities. This effect was discussed previously by Boade for pressed copper powders [15,16], and is relevant to the design of 1-dimensional (non-released) plate impact experiments.

## 2. Materials and Methods

### 2.1. Materials

The concepts discussed above will be used to interpret results obtained for the following materials: solid and porous (foamed) polyurethane [6,13,47,48,57,58], Epon- and Jeffamine-based epoxies [59,60], carbon fiber-filled phenolic (CP) and cyanate ester (CE) composites [5,60,61], and filled polydimethylsiloxane foam (SX358) [62,63]. These materials have been studied extensively over the last decade at Los Alamos National Laboratory, and represent solid-density unfilled polymers, polymer-filler composites, and two types of polymer foams. Table 1 provides their common names, chemical compositions by weight, and initial densities, which cover a range in the case of foams [5,6,63,64,65]. We define porosity level as $100\times V_{0}/V_{00}$, where $V_{0}$ is the initial volume of solid-density material and $V_{00}$ is that of the associated foam.

Polyurethane foams were prepared by condensation polymerization of the isocyanate moiety in methylene diphenyl diisocyanate (PMDI) with a hydroxyl group, yielding a molar composition (normalized to hydrogen) of C${}_{0.79}$HO${}_{0.27}$N${}_{0.08}$. The final architecture was that of an open cell foam with pore diameters bimodally distributed around 70 and 240 μm. Representative X-ray computed tomographs for the intermediate density polyurethane foams are shown in Figure 4 [6]. New experimental data at intermediate densities $\rho_{0}$ = 0.867, 0.626, 0.488, and 0.348 g/cm${}^{3}$ are combined with historical data from the LASL Compendium (pedigree unspecified) [31] below.

SX358 foam samples were prepared by adding 5 wt % Sn-octanoate catalyst to a resin mixture of polydimethylsiloxane (PDMS), polymethylhydrosilance (PMHS), tetrapropylsilicate (TPS), diphenylmethylsilane (DPMS), and diatomaceous earth filler, Figure 5. The resin was cured at room temperature in a flat sheet mold, and the porous network was formed by the evolution of hydrogen during curing and cross-linking. The foam was post-cured at 100–120 °C for several hours. SX358 foams were prepared over a range of initial densities from ∼0.4 to 1.12 g/cm${}^{3}$. H${}_{2}$(g) was evolved as part of the cross-linking process, producing an open cell, stochastic structure with pore diameters ranging from 10 s of μm to mm in diameter [65,66]. An X-ray computed tomograph is shown in Figure 6 for $\rho_{0}=0.4$ g/cm${}^{3}$.

### 2.2. Gas Gun-Driven Plate Impact Experiments

Most modern shockwave compression experiments are based on projectile impact driven by light gas guns or direct laser drive (not discussed here). Smooth bore launch tube gas and power-driven guns are widely used for imparting well-defined, flat-topped shock waves with 100 s ns to several-μs shock durations. An additional advantage of gas gun experiments is that complex loading conditions such as shock-release, double shock, or ramp loading can readily be generated using tailored impactors. The Shock and Detonation Physics group at LANL houses a two-stage light gas gun (50 mm launch tube bore) [67], a high-performance powder-driven two-stage gun (28 mm launch tube bore) [68], and a 72 mm launch tube single-stage light gas gun [69]. Achievable projectile velocities collectively span the range from ≈0.1 to 7.5 km/s.

Data discussed in Section 3 were generated using several different types of plate impact experiment. In the front surface impact (FSI) or “reverse ballistic” configuration, a sample 2–6 mm thick was mounted onto the front of a polycarbonate projectile and impacted into a standard window of single-crystal LiF ([100]); a schematic is provided in Figure 7A. The state of the material at impact was measured directly, and particle velocity at the impact interface ($u_{\mathrm{int}}$) was monitored with dual velocity-per-fringe (vpf) interferometers (VISARs) [70,71]. More recent velocimetric techniques such as photonic Doppler velocimetry (PDV) [72] were also used to measure shock velocities and wave profiles at a windowed interface. These experiments typically were performed with the 50 mm launch tube light gas gun.

In the FSI experiments, measured velocities of the sample-LiF interface were combined with projectile velocities ($u_{\mathrm{pr}}$) to obtain final (*P*, *u*) states in the sample. We calculated final pressures in the initially unshocked ($P_{0}\approx 0$) LiF by substituting its Hugoniot ($\rho_{0}=2.64$ g/cm${}^{3}$, $c_{0,\mathrm{LiF}}=5.15$ mm/μs, $s_{\mathrm{LiF}}=1.353$) [73] into the Rankine-Hugoniot relation for conservation of momentum, then equating its particle velocity with that of the interface,
(13)$$P_{\mathrm{LiF}}=\rho_{0,\mathrm{LiF}}U_{\mathrm{LiF}}u_{\mathrm{LiF}}=\rho_{0,\mathrm{LiF}}\left(c_{0,\mathrm{LiF}}+s_{\mathrm{LiF}}u_{\mathrm{LiF}}\right)u_{\mathrm{LiF}}=\rho_{0,\mathrm{LiF}}c_{0,\mathrm{LiF}}u_{\mathrm{int}}+s_{\mathrm{LiF}}u_{\mathrm{LiF}}u_{\mathrm{int}}^{2}.$$
Pressures in the sample were determined by impedance matching to the LiF,
(14)$$P_{\mathrm{LiF}}=P_{\mathrm{sample}}=P.$$
and specification of the final shocked states was complete upon relating sample particle velocities to those measured for the projectile and at the interface
(15)$$u_{\mathrm{pr}}-u_{\mathrm{int}}=u_{\mathrm{sample}}=u.$$
This procedure is known as impedance matching. From this (*P*, *u*) combination, shock velocities *U* were found by additional application of (Equation 13) to the sample only, and specific volumes $V_{H}$ (or densities, $\rho_{H}$) by Equation (Equation 1).

The second type of experiment employed the “top-hat” or transmission configuration shown in Figure 7B. In most experiments, a symmetric impact condition was created by launching either an oxygen-free, high-conductivity (OFHC) Copper or 6061 Aluminum disk into a drive plate of the same material, which was then was backed by a disk of the sample and a thick (9–12.6 mm) rear window of oriented [100] LiF. Shock transit times through the sample were measured independently using 3 each of PDV and VISAR probes—two on the rear surface of the drive plate and one on the rear-windowed interface of the sample. The measured transit time and initial sample thickness were used to calculate an initial shock velocity U=Δx/Δt, which was then corrected for impact tilt based on the cross-timed PDV and VISAR diagnostics. Projectile velocities were measured several microseconds prior to impact either by a collimated PDV probe or the sequential “cut-off” of 4 laser diodes at the launch tube exit aperture. Errors in measured projectile velocities were typically < 0.1%.

Shock velocities measured in the transmission experiments were used in combination with measured projectile velocities and the Hugoniots of OFHC-Cu ($\rho_{0}=8.93$ g/cm${}^{3}$, $c_{0}=3.94$ mm/μs, s=1.489) [42] or 6061 Al ($\rho_{0}=2.703$ g/cm${}^{3}$, $c_{0}=5.35$ mm/μs, s=1.34) [31] to calculate *u* via the impedance matching procedure described above. Shocked states in the sample were found from the intersection of its Rayleigh line ($m=\rho_{0}U$) with the Hugoniot of the projectile centered at the projectile velocity ($u=u_{\mathrm{pr}}$), assuming coincidence of the Hugoniot and isentrope in this regime. The remaining Rankine-Hugoniot variables ($\rho_{H}$, $P_{H}$, and $E_{H}$) followed from the conservation relations.

A modified form of the top-hat configuration was also used to obtain shocked states for up to 4 foam samples in a single experiment [6,63]. In this “multi-slug” target, the samples were affixed to the rear surface of the drive plate and windowed using PMMA. A 5–6 μm Al foil was glued onto the PMMA window to provide a reflective surface for detecting shock wave arrival following transit through the foam. This setup permitted collection of multiple data in a single experiment, and all with the same impact condition. Figure 8 shows a diagram of the multi-slug configuration that was applied to polyurethane and SX358 [6,63].

A final configuration was developed specifically to obtain deep-release pathways following shock compression above the threshold for chemical decomposition [74,75]. A polymer sample was affixed to the front of a projectile and driven into an oriented [100] single-crystal LiF window, similar to the FSI configuration. In this case, however, the sample was backed by a low-density glass microballoon ($\rho_{0}$ = 0.54 g/cm${}^{3}$) or polyurea foam ($\rho_{0}$ = 0.25 g/cm${}^{3}$), such that it released isentropically upon arrival of rarefaction waves from the sample-backer interface. The window diameter and thickness were both nominally 25.4 mm, and the impact surface of the window was vapor-deposited with a thin layer of Al to serve as a reflector for velocimetry measurements. Figure 9 shows the experimental configuration used for deep-release experiments on epoxy and polyethylene [75], including the probe positions for PDV and VISAR velocimetry measurements.

### 2.3. Equations of State

#### 2.3.1. Global

The EOS of polymers can be described with a variety of models, differing widely in their quality and level of detail. For simplicity we will focus on a very flexible framework known as SESAME. SESAME is also a database of tabular EOS (most of which are based on the SESAME decomposition of Equation (Equation 16)) maintained and distributed by Los Alamos National Laboratory. We will distinguish library entries from the framework by the use of all caps followed by an entry number, such as SESAME 7603 below. The total Helmholtz free energy is decomposed as [76]
(16)$$F\left(\rho ,T\right)=\phi \left(\rho \right)+F_{\mathrm{ion}}\left(\rho ,T\right)+F_{\mathrm{elec}}\left(\rho ,T\right),$$
where analogous sums apply to the internal energy and pressure. The first term represents the energy of a static lattice at zero temperature (SESAME was developed with metals in mind), with all ions and electrons in their ground state; it is often referred to as the cold curve. The second term represents the free energy of ionic excitations in the ground electronic state, and the last that of electronic excitations in the ionic ground state. The first and second terms are coupled through the Grüneisen parameter,
(17)$$\Gamma =V{\left(\frac{\partial P}{\partial E}\right)}_{V}=-{\left(\frac{\partial \ln T}{\partial \ln V}\right)}_{S},$$
and coupling of the latter two is neglected (the Born-Oppenheimer approximation [77]). The actual models employed for each are flexible, although some generalizations can be made.

For simplicity and computational speed, $F_{\mathrm{elec}}$ typically is the Thomas-Fermi or Thomas-Fermi- Dirac model [78,79,80]. These guarantee the correct physics in the high temperature limit, but can be quite inaccurate at temperatures of ≲2000 K. This is of little relevance to polymers under shock loading, however, because the electronic contributions to the energy and pressure are ≲1% up to well over a Mb in pressure on their principal Hugoniots. This is well beyond the point at which they chemically decompose, calling for an entirely different theoretical treatment described in the following section.

Most ionic models are some variation on that of Einstein or Debye [81], supplemented by (somewhat arbitrary) forms for interpolating the specific heat from its classical limit of 3R/mol at melt to the ideal gas value of 1.5R/mol at high temperature [82,83], where R is the gas constant. A feature distinguishing polymers and molecular solids from most metals and oxides is the need for multiple characteristic temperatures due to presence of both relatively low-frequency phonons and high-frequency vibrons. One means of dealing with this is through combination of Einstein models for “group” modes (vibrons) and Debye oscillators of varying dimension for “skeletal” modes. Use of variable dimension in the latter case permits treatment of chain-like, sheet-like, and 3-D crystalline vibrations [7].

Several different cold curves have been used for polymers in compression, the Tait form being particularly popular [2,84]. Cold curves for non-polymers often are calibrated to room temperature compression in a diamond anvil cell [30], where volume is measured by diffraction and pressure by reference to a standard. The former requires a high degree of crystallinity, which many polymers lack. Another common source for cold curves in metals is density functional theory calculations [85] which, again, are non-trivial for polymers due to their complex chemical structure. For these reasons, probably the most common basis historically [31] for building polymer cold curves has been by fitting to shock data [86]. Because the final term in (Equation 16) is small for modest compressions (i.e., where there are plate impact data), the total pressure and energy on the Hugoniot may be approximated as the sum of the first two terms only. If a characteristic temperature or temperatures is assumed, the cold contribution can be obtained by subtraction of the ionic contribution from the Hugoniot. Depending on the details, this procedure can have significant consequences for use of such an EOS in hydrodynamic simulation (see Section 3.5).

#### 2.3.2. Thermochemical

The EOS of shock-driven reaction products discussed below were based on thermochemical modeling [87], where full thermodynamic equilibrium of chemically distinct atomic, molecular, or solid components is assumed. The first two component types are in the fluid phase (all non-solids are well above their critical points at the relevant conditions), and their free energies are decomposed into ideal and non-ideal contributions [81]. The former are treated exactly, based on standard decomposition of the partition function into a product of vibrational (harmonic oscillator), rotational (rigid-rotor), translational, and electronic contributions. Each of these, in turn, takes gas phase vibrational frequencies, rotational constants, and electronic excitation levels as input parameters [88].

Non-ideal contributions to the free energy of fluid components were described by soft-sphere perturbation theory [89] based on exponential-6 pair potentials [90]. Such potentials have the form
(18)$$\phi \left(r\right)=\epsilon \left[\frac{6}{\alpha -6}e^{\alpha (1-r/r_{s})}-\frac{\alpha }{\alpha -6}{\left(\frac{r_{s}}{r}\right)}^{6}\right],$$
where α, $r_{s}$, and ϵ are parameters characteristic of a given fluid constituent. Please note that (Equation 18) lacks angular-dependence, meaning that even a constituent with interactions so directional as those of H${}_{2}$O is treated as isotropic. In addition to procedures designed to extract effective spherical potentials from anisotropic ones [91], the quality of this approximation will improve with temperature. Because shock-driven decomposition always involves temperatures of $T≳10^{3}$ K, some of this anisotropy is “washed out” by thermal agitation. The Gibbs free energies of all $N_{fl}$ fluids and $N_{s}$ solid components were combined into that of the mixture via
(19)$$G_{\mathrm{mixture}}\left(P,T\right)=\sum_{i=1}^{N_{fl}}x_{i}G_{i}^{\left(0\right)}\left(P,T\right)+RT\sum_{i=1}^{N_{fl}}x_{i}\ln x_{i}+\sum_{i=1}^{N_{s}}x_{i}G_{i}\left(P,T\right),$$
where the $G_{i}^{\left(0\right)}$ are free energies of fluid components *in isolation* at the same (P,T) state as that of the mixture. The middle sum represents the free energy of mixing - here assumed to be ideal - meaning that all activity coefficients are unity and therefore that mixing is a purely entropic phenomenon [92]. While obviously a crude approximation, its virtues are computational simplicity and lack of need for cross-potentials. Equation Equation 19 clearly is not unique, and other prescriptions [93,94] have been proposed. Solids contribute no mixing term, as their constituent atoms are assumed to be distinguishable from those of fluid particles. The mixture free energy is minimized as a function of $x_{i}$, subject to the constraints of stoichiometric balance and that $x_{i}\geq 0$ for all *i*.

## 3. Results and Discussion

### 3.1. Polymer Shock Data and Evidence of Decomposition

The Los Alamos Shock Compendium [31] summarizes Hugoniot data from over 5000 experiments, and the subset of polymer data were republished separately in a later report by Carter and Marsh (CM) [36]. The CM report is quite comprehensive, including results for many important polymers including polyethylene (PE), polyvinyl chloride (PVC), polytetrafluroethylene (PTFE, often Teflon^®^), polychlorotrifluroethylene (PCTFE), polymethylmethacrylate (PMMA), polystyrene, epoxy, and polyurethane. CM also discuss polymers’ propensity to undergo chemical reaction under shock loading, even providing a table of threshold pressures and degrees of volume collapse entailed (a small portion of which is reproduced in Table 2). The Lawrence Livermore National Laboratory (LLNL) [38] and Russian compendia [95] contain numerous polymer entries as well.

Data for *all* of the polymers recorded in CM display structure in their principal Hugoniots, although its degree varies widely. This structure manifests itself as a volume collapse in the P−V or P−ρ plane, or as what appear to be three straight line segments of different slope in U−u; the degree of volume collapse correlates with the length of the middle line segment. These features consistently appear at $P_{H}∼25$ GPa or u∼3 km/s. The extent of this feature can be at least qualitatively correlated with chemical structure, as illustrated in Figure 10. A single PE chain is almost entirely backbone, and the absence of pendant side chain groups means it can pack quite efficiently. When shock temperatures are sufficient to destroy the long-range order of the matrix, the amount of volume “freed up” is small relative to matrices with more excluded volume, such as polystyrene.

The most compelling evidence that this structure is caused by chemical decomposition, as opposed to some form of phase transition or additional consolidation, was provided in a set of experiments performed at LANL in the 1980s. Samples of PE [18] and PTFE [19] were exposed to single-shock, Mach compression waves in heavily confined and hermetically sealed capsules that enabled product recovery. For PE shocked to 20 GPa, the recovered sample was entirely solid PE. For PE shocked to 28–40 GPa, *no* PE was recovered; rather, the products were almost entirely methane and hydrogen gas and carbon soot that was neither graphite nor diamond.

Similar results were reported for PTFE, but the PE case is particularly notable due to it representing an extreme: because its chain structure is so clean, its volume collapse upon reaction is negligible and there is only a subtle change in slope in U−u. One would expect any accompanying temperature rise to be small and, indeed, preliminary calculations indicate that it actually *cools* upon decomposition [96]. The fact that only full decomposition products are recovered above the cusp would at least suggest *a fortiori* that this be the case for other polymers in which the volume collapse is larger.

### 3.2. Product Temperatures and Compositions

One of the most difficult quantities to measure in a dynamic experiment is temperature, making theoretical estimates particularly valuable. Thermochemical results for four different solid-density polymers and four foams (all polyurethane) are shown in Figure 11, where several features are worthy of note. The temperature increases upon reaction in all three cases, although we predict they drop in SX358 (not shown) and in PE, as already noted. This suggests that at least some polymers decompose exothermically under shock loading. The temperature rise due to reaction at constant pressure varies considerably, from <1% in epoxy to well over 100% in solid polyurethane. Foam product temperatures are a good bit higher than that for solid density at the same pressure, and their slope increases with initial porosity.

An even more difficult quantity to measure is chemical composition, and existing means for doing so are highly indirect [97]. In addition to it providing a more realistic representation of a reacting material, one of the advantages of thermochemical modeling is that it provides some physical basis for predicting this feature. Because they are based purely on thermodynamics, thermochemical compositions approximate those in the infinite time and bulk matter limits, the applicability of the former (in particular) to O(μs) experiments being not at all obvious. Figure 12 displays compositions for full-density polyurethane (left), epoxy (center), and 70% porous polyurethane (right) as functions of pressure on their Hugoniot. In each case we have excluded some minor constituents never present at >2% and included others for the sake of consistency in the presentation. Variations in epoxy composition with increasing pressure are well described by the simple equation
(20)$${\mathrm{CH}}_{4}\to C+2H_{2},$$
with water and ammonia serving largely as spectators. Polyurethane compositions show even less variation as a function of pressure, and the primary difference between full-density and foam results is the replacement of methane with hydrogen. This is to be expected, in that higher temperatures (see Figure 11) enhance the role of entropy, which is maximized by increasing moles of fluid. In each of the three cases—and we find this to be the case in general—the products as a whole are dominated by solid carbon and water.

### 3.3. Porosity and Reaction Thresholds

As described in the previous section, most full-density polymers react at ∼25 GPa on their principal Hugoniots. However, as described in Section 1.1.3, the amount of P−V work performed on the material upon loading increases dramatically with initial porosity. This has the effect of reducing the pressure needed to input a given energy, meaning also that shock heating is much greater at a given shock pressure. Standard reaction rate laws such as Arrhenius are strongly temperature-dependent, so perhaps it is not surprising that the pressures needed to observe shock-driven decomposition on the timescale of dynamic experiments drops dramatically as a function of initial porosity. By calibrating reactant and product EOS to full-density material, adding a *P*-α compaction model [8,17,52,98] to account for porosity in the unreacted material (this involved only one adjustable parameter), and adjusting the initial density as required to calculate the porous Hugoniots, we were able to clearly distinguish between reacted and unreacted polyurethane under shock loading [6]. By taking the threshold for reaction as the midpoint between the lowest-pressure reacted and highest-pressure unreacted points (and setting the uncertainty accordingly), we estimated this threshold as a function of initial porosity, as shown in Figure 13. Its value drops by more than an order of magnitude, from 26 GPa at full density to just over 1 GPa at 75% porosity.

### 3.4. Wave Profiles

One of the great advantages of modern velocimetric diagnostics is their ability to measure wave profiles. Older diagnostics provided mean wave speeds based on times of arrival, whereas profiles capture their full temporal evolution including reshock, release, and wave splitting. The last is particularly helpful for identifying shock-driven chemistry, and dramatic multiwave structures have been observed in conjunction with decomposition of organic liquids [99].

It was only recently that we reported the first observation of multiwave structure due to chemical reaction in a polymer, although with structure much less dramatic than that shown in Ref. [99]. Transmission experiments at shock stresses of approximately 30 to 50 GPa were performed on CP and CE, using the configuration of Figure 7B. Particle velocity profiles measured with VISAR at the rear CE/LiF windowed interface are shown in Figure 14 (offset arbitrarily in time), where a dashed line has been used to indicate the average interface particle velocity. The profiles were selected from shots with input shock stresses ranging from 29.1 to 46.8 GPa. The rounding in the shock front at 29.1 and 34.7 GPa constitutes a form of two-wave structure, as discussed above. Reaction occurred in the 2nd wave, and so its risetime provides the timescale for shock-driven decomposition. At 29.1 GPa, for example, chemical reactions transformed the composite to higher density products over a period of roughly 45 ns. The transmitted shock fronts sharpen into single waves above 40 GPa, and rounding in the front disappears within the temporal resolution of the VISAR measurement at 46.8 GPa. This indicates that transformation of the material was complete within the measured risetime of the shock, ∼1 ns as determined by VISAR and PDV. Measured Hugoniot states in the high-pressure regime (>40 GPa) lie on the products locus. The mixed phase region, in which two waves appeared, extended over a large pressure range from 25 to 40 GPa.

### 3.5. Reaction Reversibility and Hydrodynamics

Figure 15 depicts epoxy shock data alongside three different EOS drawn from the SESAME database. SESAME 7602-3 are based upon the SESAME decomposition of Section 2.3.1, whereas 97607 is a thermochemical EOS meant to describe reaction products only, as described in Section 2.3.2. The cold curves of SESAME 7602-3 (ϕ of Equation (Equation 16)) are based on fits to Hugoniot data: that used to build 7602 ignores the points in the mixed phase regime (artifacts of recording the arrival of the first wave only, see Section 1.1.2), whereas that of SESAME 7603 incorporates them and thereby retains their full structure. This structure propagates to all thermodynamic loci (e.g., isotherms, isentropes, etc.) until the temperature is high enough that the ionic and electronic contributions (which do not possess this structure) are together sufficient to mute it.

Figure 16 compares results obtained for epoxy in the experimental configuration of Figure 9. The black lines are averages of PDV 1, 2, 5, and 6, while the colored lines are based on hydrodynamic simulation using the same pair of SESAME EOS shown in Figure 15. Agreement between theory and experiment is good for the peak velocities, as one would expect given that all the EOS are in part calibrated to Hugoniot data. The point at which the release wave arrives at the interface is largely a function of the sound speed at pressure in the shocked material, differences that are highlighted in the insets. Here the thermochemical EOS performs noticeably better, reducing the error in sound speed from 16% in 7602 and 29% in 7603 to 8% in 97607. The improvement is less for the higher pressure shot shown on the right, but still clearly discernible. A more curious feature is the obvious multiwave structure seen in the simulation performed with 7603.

The origin of this multiwave structure is provided in Figure 17, where release paths have been included from shocked states above the threshold for reaction. The isentrope from 7603 retains the structure built into the cold curve, as shown in Figure 15 (right). Relaxation of the pressure back through this feature leads to wave splitting and thus “back reaction”. Because the products are treated as an entirely separate material—with a different EOS—no such feature is present in 97607. Similar results hold for 7602, where the structure due to reaction was not included as part of the cold curve. The way shock-driven transitions are incorporated into EOS representations can thus have hydrodynamic consequences, depending on how hard the material is shocked and how long it is simulated.

## 4. Conclusions

Polymers decompose chemically under shock loading of sufficient strength, which tends to be $u_{p}∼2.5$–3.5 km/s and P∼20–30 GPa on their principal Hugoniots. The reaction is accompanied by an increase in density, the extent of which varies widely and correlates at least qualitatively with initial chain structure and degree of crystallinity. We believe the products of this reaction to be those of full chemical decomposition (i.e., the polymer “blows apart”) and not additional consolidation of the original matrix, as in a polymorphic phase transition. The transition manifests itself as a cusp in the principal Hugoniot (Figure 1A) and as multiwave structure in particle velocity profiles obtained in situ (Figure 1B) or at interfaces (Figure 14).

Introduction of porosity complicates the picture largely through the significantly higher temperatures generated in the process of pore collapse. This effect substantially lowers the threshold pressure and particle velocity for shock-driven decomposition (Figure 13), even by an order of magnitude in 75% porous PMDI polyurethane. Thermal expansion due to shock heating can be so considerable that the Hugoniot becomes anomalous in the sense that final volumes actually increase with increasing input stress (Figure 3C), an effect observed also in metal foams. Detonating high explosives also expand as they react, but with exothermic heat release sufficient to drive a self-sustaining wave [54]. While preliminary indications are that some solid polymers *do* decompose exothermically (Figure 11), the degree of heat release is insufficient to compensate for the effects of volume collapse and the criterion for detonation is not satisfied. At high porosities, the data also exhibit a high degree of scatter, likely due to some combination of “hot spots” from void collapse, nearly equivalent wave velocities $U_{s}∼u_{p}$, and non-equilibrium conditions.

### Future Directions

There are several outstanding questions regarding shock-driven compression and dissociation of polymers and foams. As in the case of high explosives, in situ measurement of the product composition remains challenging experimentally, and even the best post-mortem studies are now decades old [18,19]. Our current treatment of carbon (based mostly on graphite shock data [31]) is particularly crude: as diamond at pressures P≳20 GPa, and as graphite at P≲20 GPa. X-ray-based methods are promising in their ability to penetrate the optically dense, high-pressure–temperature product mixture, and we have recently reported the evolution of carbon particle size and morphology in a detonating explosive in situ [102,103].

New in situ measurements of reactive wave profiles in polysulfone [104] demonstrate strong temperature-dependence of chemical reaction rates and complex two-wave structures such as those observed in CP and CE. These wave profiles are being used to calibrate reactive flow models for polymers, the first of their kind. However, EOS temperatures are unconstrained by experiment and most reaction rate forms depend exponentially on temperature, so a great deal of uncertainty regarding the details remains. Many interesting facets of the chemistry—mechanisms, intermediates, even the number of basic steps—are almost completely unknown. In the absence of such knowledge, there is little justification for use of anything beyond a single, global Arrhenius reaction rate.

Precise measurement of shock response in foams is also hindered by several sources of ill-quantified uncertainty. Sample heterogeneity (often of an extreme degree, see Figure 4) and high shock temperatures (Figure 11) can reduce the quality of velocimetric and embedded gauge data, as well as that of other diagnostics. Impedance matching to a high impedance impactor or drive plate can result in errors in particle velocity with measured shock velocities. In addition, it remains the case that only the initial (first) shock breakout is measured in many experiments, and wave profiles that might otherwise display temporal evolution are incapable of doing so. Advances in in situ and spatially resolved diagnostics, such as direct density measurements using proton or X-ray radiography and multiple point or line imaging velocimetry, offer the potential for reducing these errors.

## Figures and Tables

**Figure 1 polymers-11-00493-f001:**
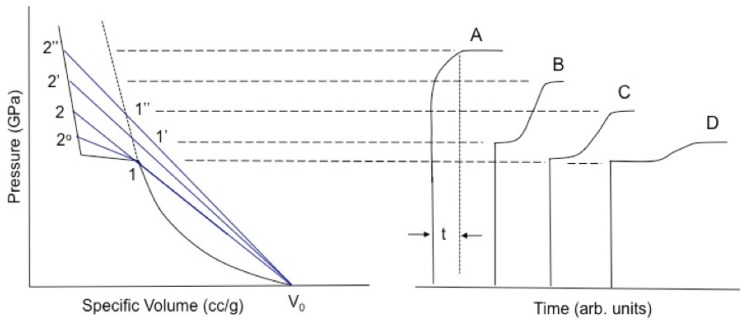
Shock-driven transitions such as chemical reaction or phase transformation that entail a reduction in volume can cause splitting of the initial wave. A transition with an onset at 1 on the principal Hugoniot will produce a waveform qualitatively similar to that of D. As the input stress is increased to 1’, 1”, etc., the temporal separation between the initial and reactive wave lessens (the plateaux on the right shorten in moving from D to B) until the transition is “overdriven” (as in A) and appears only as rounding in the initial wave. At sufficiently high input stress, even this rounding will disappear, and the transition will complete in the risetime of the lead shock. (reprinted with permission)

**Figure 2 polymers-11-00493-f002:**
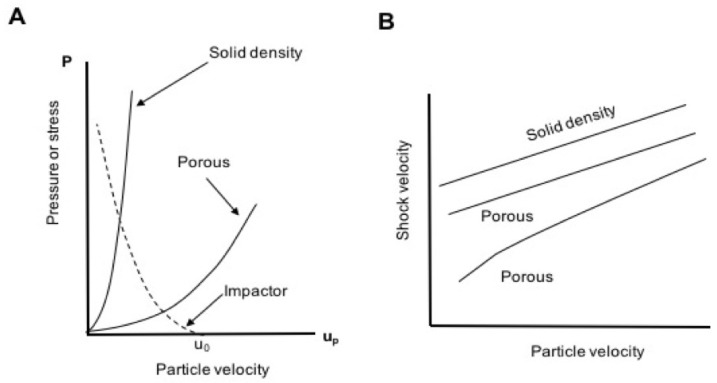
(**A**) Schematic Hugoniot curves for a solid material and its foam in the pressure-particle velocity (P−u) plane. The slope of the Rayleigh line connecting the origin to a point on either curve is $\rho_{0}U$ (see Equation (Equation 2)), also known as the *shock impedance* of the material; porosity lowers the impedance relative to that at full density. (**B**) The Hugoniot of many materials is linear in the shock velocity-particle velocity (U−u) plane. Porosity has the effect of lowering *U* at a given *u*, and can produce curvature at low *u* due to the enhanced compressibility associated with pores.

**Figure 3 polymers-11-00493-f003:**
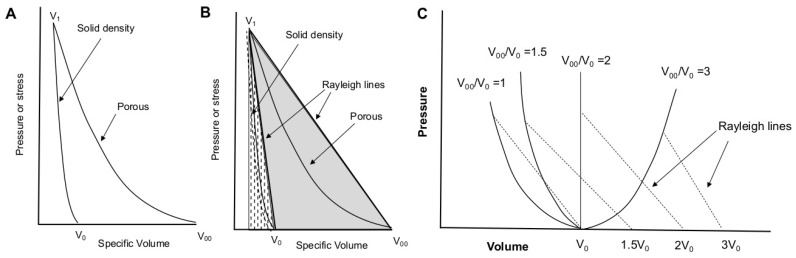
(**A**) Hugoniots for solid-density and porous material in the normal (non-anomalous) regime. The foam is compressed from initial specific volume $V_{00}$ to solid volume $V_{1}$, undergoing far greater compression than the solid material shocked from $V_{0}$ to the same final volume. (**B**) The internal energy rise upon shock loading is given by the area under the Rayleigh line, indicated by the solid gray and dotted areas for porous and solid material, respectively. The internal energy (and thus, temperature) rise is much greater in the porous case than in the solid. (**C**) As in (**A**,**B**), but showing the effect of anomalous compression, in which the porous material Hugoniot does not approach that of the solid due to shock heating.

**Figure 4 polymers-11-00493-f004:**
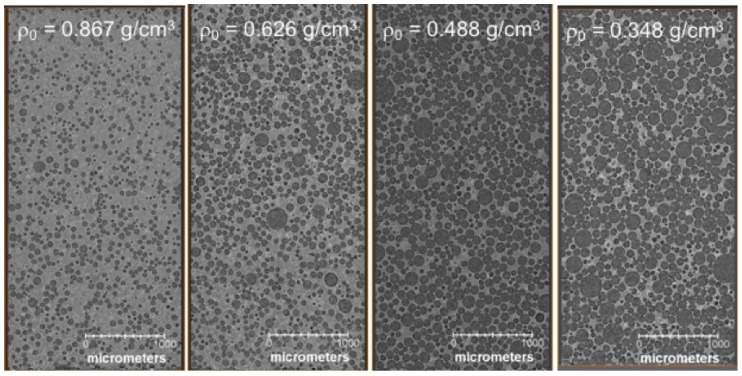
X-ray computed tomographs of PDMI-polyurethane foams with initial densities $\rho_{0}=0.867$, 0.626, 0.488 and 0.348 g/cm${}^{3}$. Taken from Ref. [6], reprinted with permission.

**Figure 5 polymers-11-00493-f005:**
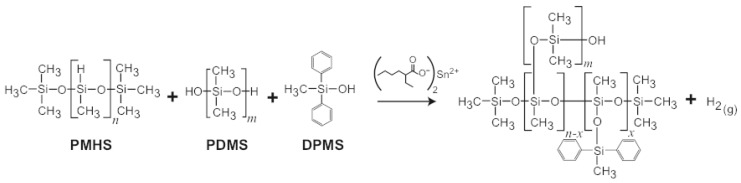
Chemical structures of starting materials, curing agent, and final SX358 foam. The H${}_{2}$(g) released during cross-linking produces the stochastic pore network of the open cell foam.

**Figure 6 polymers-11-00493-f006:**
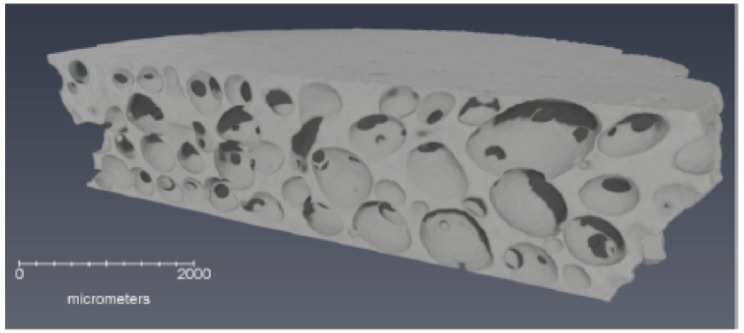
X-ray computed tomograph of SX358 foam illustrating the open cell, stochastic pore structure. Pore diameters range from 10 s of μm to mm in diameter. (image credit: Brian Patterson, Los Alamos National Laboratory)

**Figure 7 polymers-11-00493-f007:**
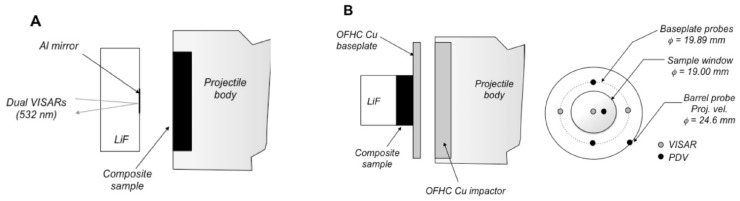
(**A**) Front surface impact geometry. Samples (in this case, a composite) were mounted to the front of a Lexan projectile, then launched into an oriented [100] single-crystal LiF window using a two-stage light gas gun. The interface particle velocity was measured using dual-velocity-per-fringe VISARs and the Doppler shift of light from a diffuse 8 kÅ Al reflector at the impact interface. (**B**) Schematic of the “top-hat” target configuration used for making shockwave transmission measurements in high-pressure plate impact experiments driven by a two-stage powder gun. Shockwave transit times are determined from the difference in shock arrival at the rear surface (sample side) of the baseplate and the sample/window (LiF) interface, as determined using multiple VISAR and PDV probes corrected for system timing and impactor tilt. Probe placements are shown at right, based on a rear-view of the target. An 8 kÅ Al reflector was coated onto the LiF window at the LiF/sample interface for the VISAR and PDV velocimetry measurements.

**Figure 8 polymers-11-00493-f008:**
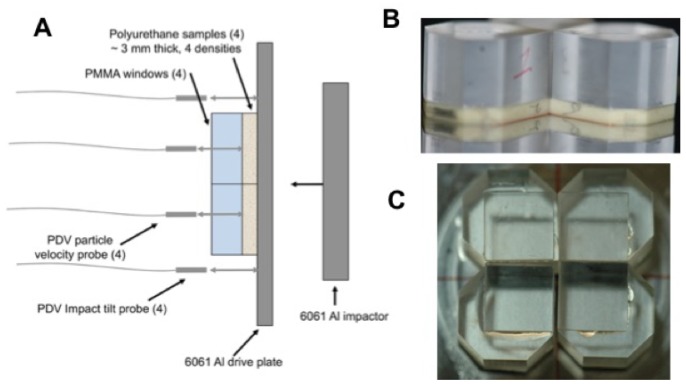
Multi-slug target configurations used at LANL large-bore two-stage gun to obtain up to 4 Hugoniot states in a single experiment. (**A**) Four foam samples with different initial densities were glued to the rear surface of a driveplate made of 6061 Al or other EOS standard material. (**B**) The samples were then windowed individually using PMMA. (**C**) A 5–6 μm Al foil was glued onto the PMMA window to provide a reflective surface for measuring shock wave arrival following transit through the foam.

**Figure 9 polymers-11-00493-f009:**
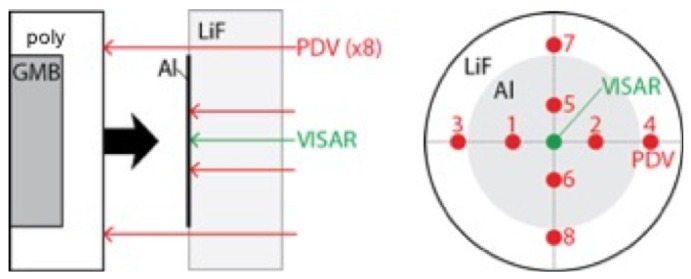
Deep-release configuration developed to access release pathways from states above the dissociation threshold on the principal Hugoniot ($P_{H}>$25 GPa). Particle velocities measured at the interface with dual VISAR and multiple points of PDV were used to characterize the shock state and release isentrope.

**Figure 10 polymers-11-00493-f010:**
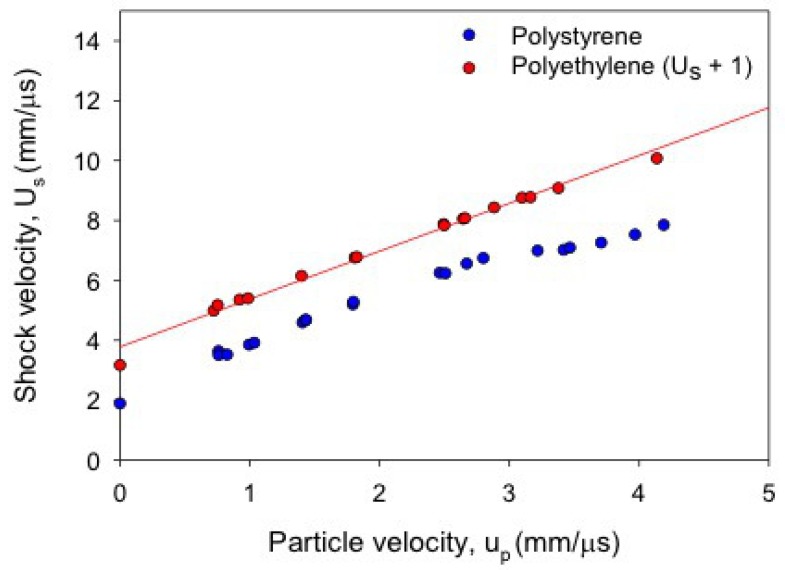
Shock Hugoniot data in the U−u plane for polyethylene (red, offset by Us + 1) and polystyrene (blue). The structure around u=3 km/s is greater in polystyrene, which undergoes a larger volume collapse.

**Figure 11 polymers-11-00493-f011:**
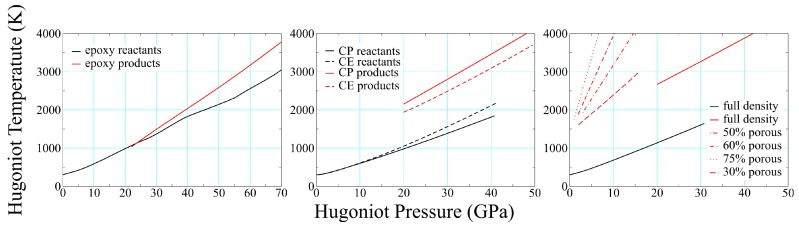
Reactant and product temperatures for epoxy (**left**), CP and CE composites (**center**), and both porous and solid-density PMDI polyurethane (**right**). All reactant curves are black, product curves are red; only the full-density reactant curve is shown for polyurethane.

**Figure 12 polymers-11-00493-f012:**
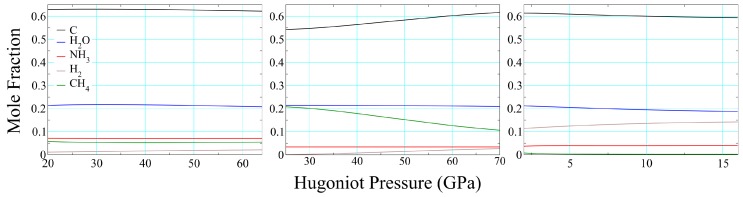
Chemical composition of the product mixture along the products Hugoniot of solid PMDI polyurethane (**left**), epoxy (**center**) and 75% porous PMDI polyurethane (**right**).

**Figure 13 polymers-11-00493-f013:**
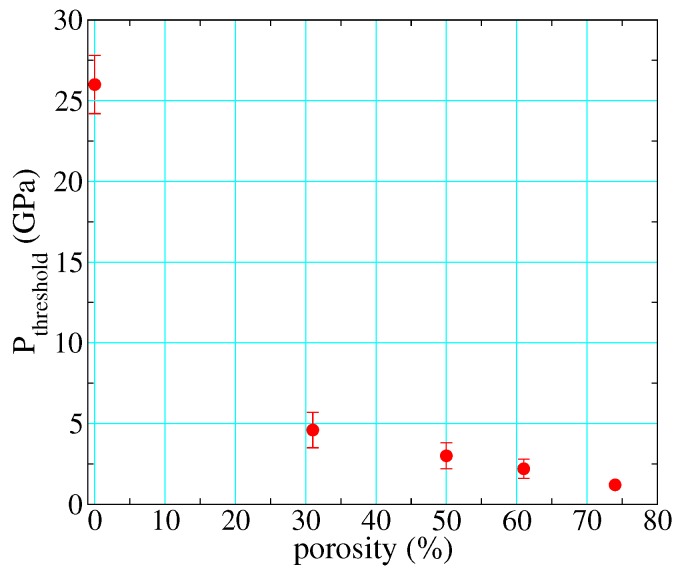
Threshold pressure for shock-driven decomposition of PMDI polyurethane along its principal Hugoniot, as a function of initial porosity.

**Figure 14 polymers-11-00493-f014:**
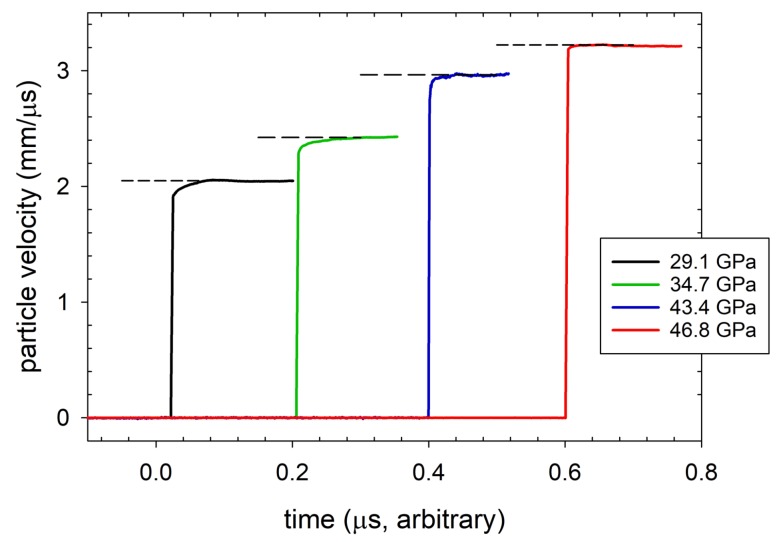
Transmitted particle velocity wave profiles measured at the windowed sample/LiF interface in selected top-hat experiments performed on the CE composite. The profiles span pressures from 29.1 to 46.8 GPa. Average interface particle velocities for each experiment are indicated in the Figure by the dashed lines. (reprinted with permission)

**Figure 15 polymers-11-00493-f015:**
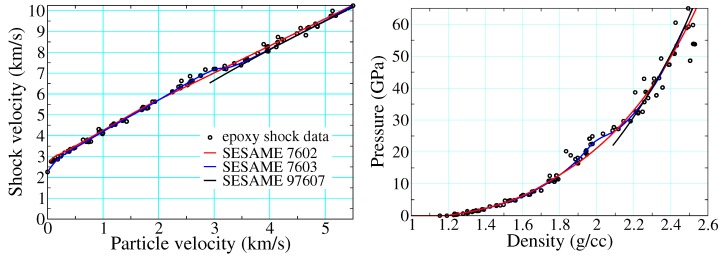
Hugoniot data for epoxy taken from Refs. [31,38,100,101], as compared with those calculated using three different numbered equations of state.

**Figure 16 polymers-11-00493-f016:**
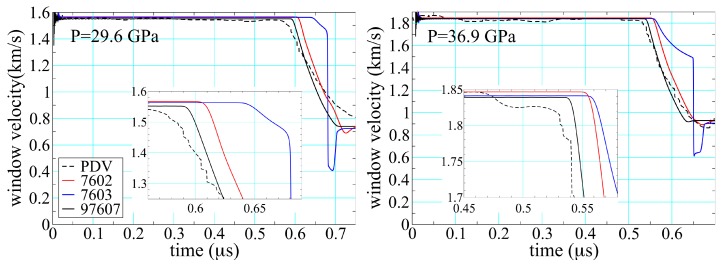
Particle velocity at the sample/window interface in deep-release experiments performed on epoxy; see Figure 9 for the experimental configuration. The average experimental PDV trace is dotted, while solid lines are results of hydrodynamic simulation performed with the SESAME EOS indicated and have the same color correspondence as those in Figure 15.

**Figure 17 polymers-11-00493-f017:**
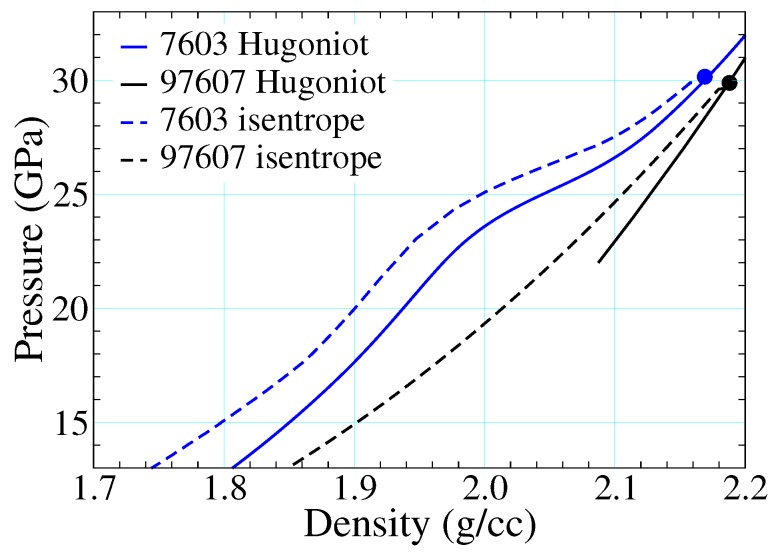
Principal Hugoniots and release isentropes (originating from a shocked state above the threshold for chemical decomposition) as read from SESAME EOS 7603 and 97607. Please note that the structure built into the cold curve of 7603 (see Figure 15, right) is retained also in the isentrope.

**Table 1 polymers-11-00493-t001:** Polymers and foams discussed in the Results section. All materials were manufactured by Department of Energy laboratories. PMDI = Diphenylmethyl diisocyanate, MW = molecular weight, PDMS = polydimethylsiloxane, PMHS = Polymethylhydrosiloxane.

Material Name	Chemical Formulation (wt %)	Initial Densities (g/cm${}^{3}$)
epoxy	Epon 828 resin (70)	1.154
	Jeffamine T-403 curing agent (30)	
polyurethane	PMDI-based polyurethane (100)	1.264, 0.867, 0.626,
(solid and porous)		0.451, 0.351
carbon phenolic	Chopped carbon fibers (56.37)	1.550–1.555
	phenolic polymer resin (35)	
	graphite powder (7.75)	
	Lithium stearate (<1)	
carbon cyanate ester	carbon fibers (68.5)	1.555–1.556
	cyanate ester resin (31.5)	
SX358 foams	high MW PDMS (43.36)	0.400–0.500
	low MW PDMS (17.19)	
	diatomaceous earth (15)	
	medium MW PDMS (14.45)	
	diphenylmethylsilanol (5)	
	PMHS (3)	
	tetrapropylorthosilicate (2)	

**Table 2 polymers-11-00493-t002:** Representative polymers, their threshold pressure for decomposition on their principal Hugoniot, and the percentage volume change upon decomposition. As taken from Ref. [36].

Material Name	$P_{\mathrm{threshold}}$ (GPa)	$\Delta V_{\mathrm{tr}}/V$ (%)
epoxy	23.1	3.9
PMMA	26.2	3.4
PTFE	41.6	1.1
PE (linear)	24.7	0.4
polycarbonate	20.0	11.4
phenolic	23.2	6.7
polysulfone	18.5	12.9
polyurethane	21.7	7.3

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
