# Peer review of "Shock-Driven Decomposition of Polymers and Polymeric Foams"

_polymers, 2019, doi:10.3390/polym11030493_

Round 1
Reviewer 1 Report
The subject of this manuscript covers the change in bahavior of polymers at around 25 GPa, presenting evidence that this may be due to chemical decomposition rather than a structural rearrangement. As such, it is of considerable interest to the shock community and I would recommend that it be accepted for publication.
However, on reading the manuscript I have the impression that it was submitted in a hurry with little or no proof reading prior to submission. Some of the more obvious errors/typos etc...
The text on page 3 to page 4 (lines 65-83) mention figures 2a and 2b, which from context appear to actually refer to figures 3a and b. Could the authors sort this out?
Caption to figure 3. "...The release isentrope is also shown in the figure as the dashed line". I can see no dashed line in my copy.
Page 4, line 89. I believe that the authors have mistakenly failed to insert the relevant reference (see citekrupnikov, fickett). From further on in the text I believe this to be reference 51 (which is missing from the reference list). I think the same mistake has been repeated in the figure caption for figure 4.
Page 10, line 165. Include the values of c0 and S for 6061 aluminum (not ?? mm/us and ??). Marsh gives these values of 5.35 mm/us and 1.34 respectively.
Page 19, line 386. Add the reference number to Morris et al.
References. Many of these are incomplete (or in the case of 51 missing). Could the authors go through this list to correct?
Equations 7 and 8, page 2. The authors comment that the value of c0 , i.e. the zero particle velocity intercept with shock speed (Eq 7) is related to the isentropic bulk modulus (Eq 8). Whilst this is generally true (and indeed the authors qualify this with the term many), in most polymers, the low pressure value c0 of US=co+Sup is actually higher than the bulk sound speed determined ultrasonically (see for example Carter and Marsh and many other references). Whilst this is significantly below the stresses of interest to this manuscript, it is still an important charactersitic of the shock response of polymers. Could the authors add a brief paragraph discussing this?
Author Response
See uploaded document.

Reviewer 2 Report
The manuscript entitled "Shock-Driven Decomposition of Polymers and Polymeric Foams” concerns shock compression of polymers results in dissociation with the product mixture determined by the initial chemical stoichiometry of the polymer and any fillers. The authors' ideas are described well in the manuscript and interesting discussions for benchmark problems are provided. The manuscript also matches the scope of the journal. Thus, the reviewer thinks that the manuscript if suitable for the publication.
Author Response
See uploaded document.

Reviewer 3 Report
Overall this is a very interesting and well written paper providing a useful review of shock compression of polymers. The only comment from this reviewer is that the introduction is arguably a little too 'high level' - e.g. the background provided into shock physics is arguably slightly too wide-ranging given the relatively specialised nature of the target audience. However as the reader is free to push ahead this is a minor issue. Overall, the reviewer looks forwards to reading the final version of the paper in print.
Author Response
See uploaded document.
